# Reasons for non-attendance to cervical cancer screening and acceptability of HPV self-sampling among Bruneian women: A cross-sectional study

Liling Chaw[1]*, Shirley H. F. Lee[1], Nurul Iffah Hazwani Ja'afar[1], Edwin Lim[2], Roslin Sharbawi[3]

1 PAPRSB Institute of Health Sciences, Gadong, Brunei Darussalam, 2 Histopathology Department, RIPAS Hospital, Ministry of Health, Bandar Seri Begawan, Brunei Darussalam, 3 Community Maternal Health Service, Department of Health Services, Ministry of Health, Bandar Seri Begawan, Brunei Darussalam

* liling.chaw@ubd.edu.bn

**Data Availability Statement:** All relevant data are within the paper and its Supporting Information files.

## Abstract

### Objective

Uptake for cervical cancer screening remains well below the 80% target as recommended by Brunei's National Cervical Cancer Prevention and Control plan. We conducted a pilot study to determine the reasons for non-attendance and explore their acceptance of human papillomavirus (HPV) self-sampling as an alternative to the Pap test.

### Methods

A cross-sectional study was conducted at a primary healthcare center in Brunei, from January to December 2019. We recruited screening non-attendees, defined as women who were eligible for Pap test but who either never, or did not have one within the recommended screening interval of 3 years. This recruitment was done conveniently among women attending outpatient care and/or child health services at the primary healthcare center. Participants were first asked to complete a self-administered paper-based questionnaire on their reasons for screening non-attendance, and then invited for HPV self-sampling. Among those who agreed to participate in HPV self-sampling, they were asked to complete a second questionnaire on the self-sampling procedure and their samples were tested for high-risk HPV (hr-HPV). Results were analyzed using descriptive and inferential statistics.

### Result

We enrolled 174 screening non-attendees, out of which 97 (55.7%) also participated in HPV self-sampling. The main reasons for not attending Pap test screening were fear of bad results (16.1%, n = 28); embarrassment (14.9%, n = 26) and lack of time due to home commitments (10.3%, n = 18). When compared to those who agreed to participate in HPV self-sampling, those who declined were significantly older (p = 0.002) and less likely to agree that they are susceptible to cervical cancer (p = 0.023). They preferred to receive Pap test-

**Funding:** Edwin Lim has acquired funding from Ministry of Health, Brunei Darussalam. The funders had no role in study design, data collection and analysis, decision to publish, or preparation of the manuscript.

**Competing interests:** The authors have declared that no competing interests exist.

related information from healthcare workers (59.0%, n = 155), social messaging platforms (28.7%, n = 51) and social media (26.4%, n = 47). HPV self-sampling kits were positively received among the 97 participants, where > 90% agreed on its ease and convenience. Nine (9.3%) tested positive for hr-HPV, out of which eight were non-16/18 HPV genotypes.

## Conclusion

Our findings suggest that promoting awareness on cervical cancer, clarifying any misconceptions of Pap test results, and highlighting that the disease is preventable and that early detection through screening can facilitate successful treatment would help increase screening uptake among Bruneian non-attendees. Response to HPV self-sampling was highly positive, suggesting the possibility of implementing this strategy in the local setting. Our high detection of non-16/18 HPV genotypes suggest high prevalence of other hr-HPV genotypes in Brunei. Larger studies should be conducted to further validate our findings.

## Introduction

Cervical cancer is highly preventable but still remains one of the most common cancers among women worldwide. Globally, an estimated 604,000 women were diagnosed with cervical cancer, and 342,000 women died from the disease in 2020 [1]. Cervical cancer screening has drastically reduced the incidence of invasive cervical cancer in countries that have implemented such screening programs [2], which traditionally involves the use of the Papanicolaou (Pap) test.

Brunei Darussalam (population 459,500) is a small Southeast Asian country with a predominant Muslim population and a crude birth rate of 15.3 per 1,000 population [3]. Within this region, it has one of the highest age-standardized incidence rate (ASR) for cervical cancer: 20.6 per 100,000 women-years in Brunei when compared to 10.5 and 7.7 per 100,000 women-years in Malaysia and Singapore, respectively [4]. Brunei has initiated an organized cervical screening program since 2009, where married or ever married Bruneian women between 20 and 65 years old were invited to attend cervical cancer screening through periodic mail invitations. Pap test is the only screening test used in the country, and currently, liquid-based cytology is being used since 2012. Women with any positive cytology result, defined as with atypical squamous cells of undetermined significance (ASC-US) or worse, are followed up with colposcopy-guided cervical biopsy to diagnose cervical intra-epithelial neoplasia (CIN). The latter refers to premalignant lesions that are mainly caused by infection with certain types of human papillomavirus (HPV) [5], and can be categorized into any one of three stages (CIN1, CIN2, or CIN3) depending on the degree of dysplasia. If untreated, either CIN2 or CIN3 (collectively referred to as CIN2+) can progress to cervical cancer.

Despite this screening service being offered free of charge, the national screening coverage rate remains low at 44% in 2018 (unpublished data). Reasons for screening non-attendance can vary across settings [6, 7], but they can be broadly categorized into two groups: practical and organizational barriers (such as forgot to schedule an appointment, work and childcare commitments) [6], and emotional barriers (such as feeling healthy, lack of time, discomfort associated with gynecologic examination embarrassment, fear of smear test, previous negative experiences and dissatisfaction with their general practitioner) [6, 8–10]. As women who do not attend screening are at increased risk of developing cervical cancer [6], it is thus important to first understand why women chose not to attend screening in the local context.

HPV DNA detection has been recommended by the World Health Organization as the primary screening test for cervical cancer as specific high-risk HPV subtypes (hr-HPV) are known to be a causative agent [1]. In particular, the use of HPV self-sampling kits was suggested to increase screening uptake particularly among screening non-attendees [11, 12], due to its ease of access (where kits could be mailed to women's homes) and also flexibility for women to perform the test by themselves. Previous studies have shown HPV self-sampling to be highly acceptable among screening non-attendees [13, 14]. In addition, HPV self-sampling results exhibit similar sensitivity and specificity compared to those from samples taken by trained professionals [15]. Repeated HPV self-sampling and testing were shown to increase screening uptake [11, 12, 16], and also resulted in at least two-fold higher detection rate of CIN2+ when compared to the Pap test [17, 18]. In the United Kingdom and Australia, early detection through an organized screening program using HPV testing as the primary screening test was shown to reduce cervical cancer morbidity and mortality [19, 20].

With Brunei's relatively high incidence of cervical cancer and low screening uptake, we conducted a pilot study to explore the reasons behind non-attendance and to assess the acceptability of HPV self-sampling as a possible alternative to the Pap test among non-attendees (specifically women who are currently not accessing or attending screening). Study findings could be used to strategize ways to improve screening uptake and provide preliminary evidence towards implementing HPV testing as the primary screening test for cervical cancer in Brunei.

## Methods

### Study design and data collection

A cross-sectional survey was conducted at the Jubli Perak Sengkurong Health Center (JPSHC), from January to December 2019. JPSHC is a government-funded primary healthcare center located at Brunei-Muara District, where the majority (69.3%) of the country's population resides. This center provides primary health care services to Mukim Sengkurong, a sub-district with about 32,000 people from various socioeconomic backgrounds.

Eligible women attending either the outpatient or child health clinic at JPHSC were conveniently recruited by triage nurses. We defined screening non-attendees as married or ever married women between 20 and 65 years old and have never undergone cervical cancer screening, or did not have one within the recommended screening interval of 3 years. We excluded women who could not comprehend Brunei-Malay or English language, were pregnant, have had total hysterectomy, or with a history of malignancies.

We implemented a two-stage recruitment procedure. In the first stage, participants were first recruited to complete a self-administered questionnaire on the reasons for screening non-attendance (Q1) onsite. Q1 consists of 19 questions on the participant's socio-demographics, reasons for not getting Pap test, attitude and knowledge on cervical cancer, and lastly, preferred sources to acquire information about Pap test. From a prepared list of 16 possible reasons for not attending the screening program, participants were also asked to select one "Major" reason (defined as the main reason) and one or more "Minor" reason (defined as other reasons for not attending the screening program). Responses for questions on their attitude and knowledge on cervical cancer were recorded using a five-point Likert scale, ranging from "Strongly Agree" to "Strongly Disagree".

In the second stage, all participants were given an envelope containing information on HPV self-sampling and an instruction leaflet on the procedure, after completing Q1. Within the following two weeks, they were contacted via telephone by a trained nurse, and those who gave verbal consent were given an appointment at JPSHC. On the day, the nurse first explained the procedure using an instructional video and answered any questions. Participants were

then given a self-sampling kit and asked to perform the procedure in the clinic. After completion, they were asked to complete the second self-administered questionnaire (Q2) on their acceptability of the self-sampling procedure. Q2 consists of 12 five-point Likert scale questions (ranging from "Strongly Agree" to "Strongly Disagree") on their experiences and opinions of the self-sampling kit.

Two separate written consents were requested from participants: one for completing Q1 and another for performing self-sampling and completing Q2.

## Questionnaires used

The two questionnaires used (Q1 and Q2) were bilingual self-administered paper-based questionnaires in the two languages commonly used in Brunei (namely, Brunei-Malay and English language). Both questionnaires were adapted from similar studies [21, 22]. They were first translated to Brunei-Malay language by native speakers, and then back-translated to check for inconsistencies in comprehension. Both questionnaires were also pre-tested on six eligible women to assess if the questions could be easily understood. Responses from pre-testing were not included in the analysis.

## Self-sample handling and laboratory testing

Swabs collected from the self-sampling kits were sent to an overseas laboratory (at BNH hospital, Thailand) for hr-HPV testing. The self-sampling device used was the Evalyn® brush from Rovers Medical Devices. Upon receipt at the testing laboratory, the dry Evalyn® brush was suspended in SurePath medium from which a sample was obtained for the identification of the presence of hr-HPV using the cobas HPV test (Roche, USA). Cobas HPV tests are automated qualitative *in vitro* tests for the detection of HPV DNA in patient specimens. The tests utilize amplification of target DNA by polymerase chain reaction (PCR) and nucleic acid hybridization for the detection of 14 hr-HPV types (namely genotypes 16, 18, and 12 pooled hr-HPV genotypes 31, 33, 35, 39, 45, 51, 52, 56, 58, 59, 66 and 68) in a single analysis. Results obtained from this test can be categorized into four groups: Negative, HPV-16 positive, HPV-18 positive, and positive for non-16/18 HPV genotypes.

## Clinical management of hr-HPV positive participants

We adopted the cytology triage strategy for HPV self-sampling participants [23]. Participants with hr-HPV positive results were invited for an immediate clinic-based cytology triage test. Those found to have negative cytology results were invited for a repeat cytology triage test after six months. Those with second negative results were returned to routine cervical screening recall. Participants with any positive cytology result (defined as with ASC-US or worse) were referred for colposcopic examination.

## Statistical analysis

Descriptive statistics was conducted to characterize the socio-demographic characteristics of the study population, their reasons for not attending screening, their attitudes and preference for information access on such screening, as well as responses from Q2. Where appropriate, Mann-Whitney and/or Fisher's exact tests were used to assess significant differences in socio-demographic characteristics, reasons for not attending screening, and attitudes between women who agreed and women declined to join HPV self-sampling. Count responses for questions with the five-point Likert scale were categorized into three categories (agree, neutral, and disagree), and those with missing values were classified as neutral. Statistical analysis was

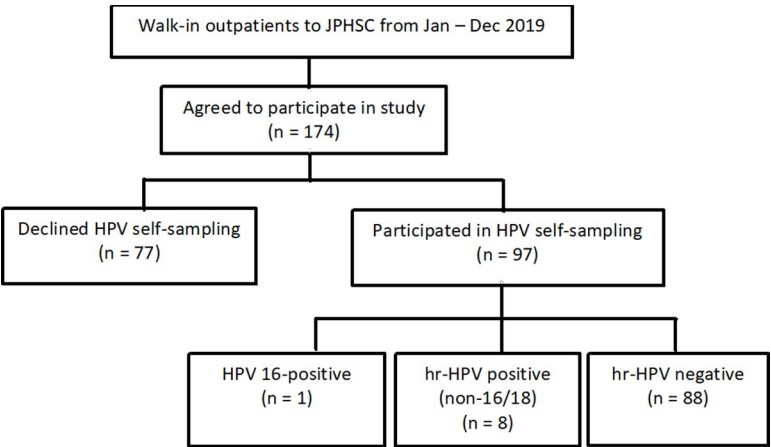

**Fig 1. Flowchart of study participants at JPHSC, Brunei (Jan–Dec 2019).**

conducted using R ver. 3.6 [24]. Ethics approval was obtained from the Medical and Health Research and Ethics Committee (MHREC), Ministry of Health, Brunei Darussalam [Reference no. MHREC/MOH/2018/9(2)].

## Results

A total of 174 eligible women were enrolled in this study from January to December 2019, out of which 97 (55.7%) also participated in HPV self-sampling (Fig 1). Their median age was 45 years, ranging between 23 and 65 years (Table 1). The participants were mainly of Malay ethnicity (92.5%, n = 161), married (90.2%, n = 157), and had $\geq$ 3 births (60.3%, n = 105). About half of the participants had their last Pap test performed between 4 and 10 years ago (52.3%, n = 91), and have never received the HPV vaccine (54.6%, n = 95). There were significant differences between those who agreed and declined to participate in HPV self-sampling: those who declined were significantly older ($p$ = 0.002) and more likely to have their last Pap test performed > 10 years ago ($p$ = 0.031).

Fig 2 and S1 Table show the responses for the major and minor reasons for not attending cervical cancer screening. The top three major reasons reported were that they were "afraid of getting a bad result" (16.1%, n = 28), "feeling embarrassed being examined by a doctor or nurse" (14.9%, n = 26), and "I can't find the time as I'm too busy at home" (10.3%, n = 18). The top three minor reasons were "feeling embarrassed being examined by a doctor or nurse" (20.7%, n = 36), "I can't find time as I'm too busy at work" (20.7%, n = 36), and "afraid of getting a bad result" (20.1%, n = 35).

When comparing top 10 major reasons for not attending screening between those who agreed and declined HPV self-sampling (Table 2), those whose major reason was "feeling embarrassed being examined by a doctor or nurse" were significantly more likely to join self-sampling ($p$ = 0.020). Also, those whose major reason was "afraid of getting a bad result" were significantly more likely to decline self-sampling ($p$ = 0.034). Among those who were employed (n = 98), about a quarter (23.5%, n = 23) reported work-related reasons as their major reason for not attending screening ("I can't find the time as I'm too busy at work" and "Difficult to get permission from employer").

While the responses vary when asked about their health and susceptibility to disease, most agreed on the benefits of undergoing Pap test (92.0%, n = 160), and that cervical cancer is a severe and potentially lethal disease (82.8%, n = 144; Table 3 and S2 Table). Those who agreed

**Table 1. Sociodemographic characteristics of the study population, including comparison between groups that agreed and declined to join HPV self-sampling.**

| Characteristics | | Total study population (n = 174) | Joined self-sampling (n = 97) | Declined self-sampling (n = 77) | p-value |
|---|---|---|---|---|---|
| | | n (%) | n (%) | n (%) | |
| Median age in years (IQR) | | 45.0 (15.25) | 41.0 (17) | 49.0 (14.5) | **0.002**\* |
| Age group (in years) | 20–24 | 3 (1.7) | 3 (100) | 0 (0.0) | **0.018**\* |
| | 25–29 | 20 (11.5) | 15 (75.0) | 5 (25.0) | |
| | 30–34 | 14 (8.1) | 10 (71.4) | 4 (28.6) | |
| | 35–39 | 28 (16.1) | 18 (64.3) | 10 (35.7) | |
| | 40–44 | 18 (10.3) | 11 (61.1) | 7 (38.9) | |
| | 45–49 | 27 (15.5) | 12 (44.4) | 15 (55.5) | |
| | 50–54 | 35 (20.1) | 15 (42.9) | 20 (57.1) | |
| | 55–59 | 14 (8.1) | 10 (71.4) | 4 (28.6) | |
| | > 60 | 13 (7.5) | 3 (23.1) | 10 (76.9) | |
| | Missing | 2 (1.1) | 0 (0.0) | 2 (100) | |
| Race | Malay | 161 (92.5) | 90 (55.9) | 71 (44.1) | 0.752 |
| | Chinese | 6 (3.5) | 4 (66.7) | 2 (33.3) | |
| | Other | 7 (4.0) | 3 (42.9) | 4 (57.1) | |
| Education level | Primary school | 16 (9.2) | 10 (62.5) | 6 (37.5) | 0.396 |
| | Secondary school | 96 (55.2) | 49 (51.0) | 47 (49.0) | |
| | College / University | 57 (32.7) | 35 (61.4) | 22 (38.6) | |
| | Missing | 5 (2.9) | 3 (60.0) | 2 (40.0) | |
| Marital status | Married | 157 (90.2) | 91 (58.0) | 66 (42.0) | 0.15 |
| | Divorced | 8 (4.6) | 2 (25.0) | 6 (75.0) | |
| | Widowed | 9 (5.2) | 4 (44.4) | 5 (55.6) | |
| Occupation | Housewife | 64 (36.8) | 39 (60.9) | 25 (39.1) | 0.087 |
| | Government employee | 67 (38.5) | 41 (61.2) | 26 (38.8) | |
| | Private employee | 31 (17.8) | 13 (41.9) | 18 (58.1) | |
| | Retired | 9 (5.2) | 3 (33.3) | 6 (66.7) | |
| | Unemployed | 1 (0.6) | 1 (100) | 0 (0.0) | |
| | Other | 2 (1.1) | 0 (0.0) | 2 (100) | |
| Monthly household income | < $500 | 27 (15.5) | 15 (55.6) | 12 (44.4) | 0.498 |
| | $500 < $999 | 27 (15.5) | 12 (44.4) | 15 (55.6) | |
| | $1000-$1999 | 40 (23.0) | 22 (55.0) | 18 (45.0) | |
| | $2000-$2999 | 19 (10.9) | 14 (73.7) | 5 (26.3) | |
| | $3000-$5000 | 24 (13.8) | 15 (62.5) | 9 (37.5) | |
| | >$5000 | 3 (1.7) | 2 (66.7) | 1 (33.3) | |
| | Missing | 34 (19.5) | 17 (50.0) | 17 (50.0) | |
| Number of births | 0 | 27 (15.5) | 17 (63.0) | 10 (37.0) | 0.103 |
| | 1 | 19 (10.9) | 14 (73.7) | 5 (26.3) | |
| | 2 | 22 (12.1) | 14 (66.7) | 7 (33.3) | |
| | 3 or more | 105 (60.3) | 51 (48.6) | 54 (51.4) | |
| | Missing | 2 (1.2) | 1 (50.0) | 1 (50.0) | |
| Last Pap test done | Never | 41 (23.6) | 29 (70.7) | 12 (29.3) | **0.031**\* |
| | 4–10 years | 91 (52.3) | 48 (52.7) | 43 (47.3) | |
| | > 10 years | 36 (20.7) | 15 (41.7) | 21 (58.3) | |
| | Missing | 6 (3.4) | 5 (83.3) | 1 (16.7) | |

(*Continued*)

**Table 1.** (Continued)

| Characteristics | | Total study population (n = 174) | Joined self-sampling (n = 97) | Declined self-sampling (n = 77) | p-value |
|---|---|---|---|---|---|
| | | n (%) | n (%) | n (%) | |
| HPV vaccination status | Unvaccinated | 95 (54.6) | 50 (52.6) | 45 (47.4) | 0.483 |
| | Fully vaccinated | 27 (15.5) | 16 (59.3) | 11 (40.7) | |
| | Partly vaccinated | 41 (23.6) | 26 (63.4) | 15 (36.6) | |
| | Missing | 11 (6.3) | 5 (45.5) | 6 (54.5) | |

IQR = Interquartile range.

that they are more susceptible to develop cervical cancer were significantly more likely to join HPV self-sampling (*p* = 0.023).

Most participants would like to obtain more information about cervical cancer screening (85.6%, n = 149). The top preferred information sources were healthcare workers (59.8%, n = 104), social messaging platforms (28.7%, n = 50) and social media (26.4%, n = 46; S1 Fig).

Among those who participated in self-sampling (55.7%, n = 97), their responses on Q2 (Table 4) were mostly positive. A majority agreed that the instructions were clear (94.8%, n = 92), that it was easy to perform the swab (93.8%, n = 91), and that it was more convenient than the Pap test (91.7%, n = 89). They also reported their confidence in correctly getting the sample (92.8%, n = 90), would prefer to use this method next time (94.8%, n = 92), and would recommend this method to other women (93.8%, n = 91). Notably, 54.6% (n = 53) still prefer a proper Pap test for their subsequent check-up.

Among the 97 samples taken, nine (9.3%) tested positive for hr-HPV: one was positive for HPV 16 and eight positive for non-HPV 16/18 HPV genotype. The HPV 16 positive case was found to have high-grade ASC-US (ASCUS-H) on the initial follow up smear, but found negative after subsequent follow up cervical biopsy. Among the other 8 non-HPV 16/18 HPV genotype positive cases: 2 had negative follow up smears, 4 were reported to have ASC-US on their initial follow up smears, but had subsequent negative follow up smears, 1 was reported to

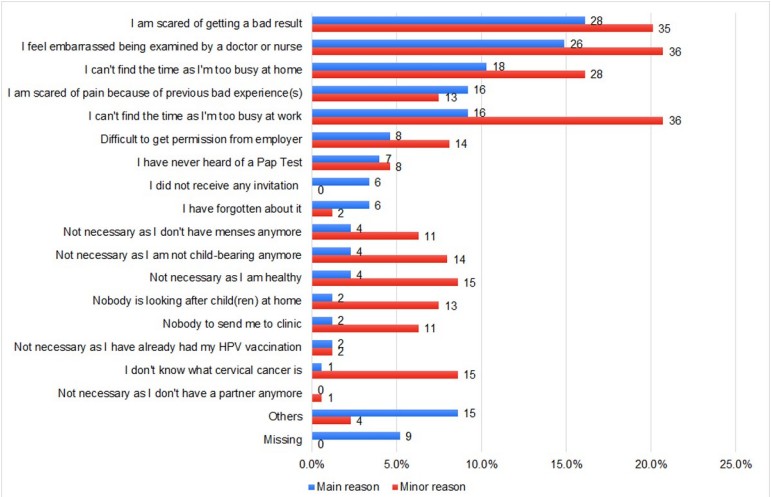

**Fig 2. Responses on their major and minor reasons for not attending cervical cancer screening, among non-attendees at JPSHC (Jan–Dec 2019).** The x-axis indicates the percentage, and the number next to each bar indicates the actual number of responses.

**Table 2. The top ten major reasons of not attending cervical cancer screening at JPSHC, Brunei (Jan–Dec 2019), with comparison between those who agreed and declined HPV self-sampling.** Responses from 146 participants (83.9% of the total study population) were included.

| No. | Top ten major reasons of screening non-attendance | Total | Joined HPV self-sampling | Declined HPV self-sampling | *p*-value |
|---|---|---|---|---|---|
|  |  | n (%) | n (%) | n (%) |  |
| 1 | I feel embarrassed being examined by a doctor or nurse | 26 (14.9) | 20 (76.9) | 6 (23.1) | **0.020**\* |
| 2 | I am scared of pain because of previous bad experience(s) | 16 (9.2) | 8 (50.0) | 8 (50.0) | 0.825 |
| 3 | I am scared of getting a bad result | 28 (16.1) | 10 (35.7) | 18 (64.3) | **0.034**\* |
| 4 | I can't find the time as I'm too busy at home | 18 (10.3) | 12 (66.7) | 6 (33.3) | 0.463 |
| 5 | I can't find the time as I'm too busy at work | 16 (9.2) | 10 (62.5) | 6 (37.5) | 0.759 |
| 6 | Difficult to get permission from employer | 8 (4.6) | 5 (62.5) | 3 (37.5) | 1 |
| 7 | I have never heard of a Pap Test | 7 (4.0) | 4 (57.1) | 3 (42.9) | 1 |
| 8 | I have forgotten about it | 6 (3.4) | 5 (83.3) | 1 (16.7) | 0.229 |
| 9 | I did not receive any invitation | 6 (3.4) | 5 (83.3) | 1 (16.7) | 0.229 |
| 10 | Others | 15 (8.6) | 7 (46.7) | 8 (53.3) | 0.492 |

have low-grade squamous intraepithelial lesion (LSIL) on her initial follow up smear, and had a subsequent negative follow up smear, and 1 was found to have CIN 3 with glandular involvement on the follow up smear and cervical biopsy, and has received treatment with a cone biopsy where excision of CIN 3 was confirmed. No significant differences were observed when comparing the sociodemographic characteristics between those who tested hr-HPV positive and those who tested hr-HPV negative (S3 Table).

## Discussion

Our study findings highlight three important points to consider for improving cervical cancer screening uptake and detection. First, our findings suggest that it is necessary to provide accurate information among Bruneian women on cervical cancer, the importance of screening and addressing any misconceptions about the Pap test. Important facts to relay include the slow development from pre-cancerous changes to cervical cancer, that pre-cancerous changes are highly treatable, and that screening will help in early detection and thus facilitate successful treatment.

**Table 3. Attitudes towards cervical cancer screening among non-attendees at JPSHC, Brunei (Jan–Dec 2019), between those who agreed and declined HPV self-sampling.**

| No. | Attitude questions | | Total study population (n = 174) | Joined HPV self-sampling (n = 97) | Declined HPV self-sampling (n = 77) | *p*-value |
|---|---|---|---|---|---|---|
|  |  |  | n (%) | n (%) | n (%) |  |
| 1 | I believe I am healthy and free of any diseases | Agree | 62 (35.6) | 34 (54.8) | 28 (45.2) | 0.984 |
|  |  | Neutral/ Disagree | 112 (64.3) | 63 (56.3) | 49 (43.7) |  |
| 2 | Having Pap test taken is beneficial for my health | Agree | 160 (92.0) | 91 (56.9) | 69 (43.1) | 0.464 |
|  |  | Neutral/ Disagree | 14 (8.0) | 6 (42.9) | 8 (57.1) |  |
| 3 | Like any women, I am susceptible to develop cervical cancer | Agree | 110 (63.2) | 69 (62.7) | 41 (37.3) | **0.023**\* |
|  |  | Neutral/ Disagree | 64 (36.8) | 28 (43.8) | 36 (56.2) |  |
| 4 | Cervical cancer can be severe and may lead to death | Agree | 144 (82.8) | 79 (54.9) | 65 (45.1) | 0.754 |
|  |  | Neutral/ Disagree | 30 (17.2) | 18 (60.0) | 12 (40.0) |  |

**Table 4. Acceptability of HPV self-sampling among non-attendees who participated in HPV self-sampling at JPSHC, Brunei (Jan–Dec 2019).** Responses from all Q2 respondents (n = 97) were included.

| No. | Self-sampling Questions | Agree | Neutral | Disagree |
|-----|-------------------------|-------|---------|----------|
|  |  | n (%) | n (%) | n (%) |
| 1 | I thought the instructions were clear | 92 (94.8) | 0 (0.0) | 5 (5.2) |
| 2 | It was easy to do the swab | 91 (93.8) | 1 (1.0) | 5 (5.2) |
| 3 | Taking the sample with the swab was painful | 12 (12.4) | 8 (8.2) | 77 (79.4) |
| 4 | Taking the sample was uncomfortable to do | 11 (11.4) | 4 (4.1) | 82 (84.5) |
| 5 | I felt embarrassed doing the self-sampling | 9 (9.3) | 1 (1.0) | 87 (89.7) |
| 6 | It was convenient to do without having to undergo a Pap Test | 89 (91.7) | 3 (3.1) | 5 (5.2) |
| 7 | I am confident I did it correctly | 90 (92.8) | 4 (4.1) | 3 (3.1) |
| 8 | I want to use this method next time | 92 (94.8) | 2 (2.1) | 3 (3.1) |
| 9 | I prefer to do this at home | 64 (66.0) | 15 (15.5) | 18 (18.5) |
| 10 | I would recommend this method to other women | 91 (93.8) | 3 (3.1) | 3 (3.1) |
| 11 | I trust that the result of this self-sampling will be accurate | 74 (76.3) | 19 (19.6) | 4 (4.1) |
| 12 | I would like to attend for a proper Pap test in clinic next time | 53 (54.6) | 23 (23.7) | 21 (21.7) |

In our study, emotional barriers (fear of unfavorable test results and embarrassment) were the most common major reasons of screening non-attendance. Also, about two-thirds of our participants cited "I am scared of getting a bad result" as their main barrier also declined to take part in HPV self-sampling, possibly due to relating abnormal Pap test results to cervical cancer diagnoses [25]. Although such barriers may play a large role at the onset of screening program, its role diminishes over time with increasing education on the benefits of screening [26]. Educational interventions could also benefit the small group of women who cited menopause, cessation of child-bearing and having had HPV vaccination as reasons for not attending screening (Fig 2 and S1 Table). We also observed that those who agree that they are susceptible to cervical cancer were significantly more likely to participate in HPV self-sampling. This suggests that perceived susceptibility could be an important factor for self-sampling participation, whereby those who do not perceive themselves as susceptible were less likely to engage in preventative behaviors [27–29]. Perceived susceptibility can be increased through education to improve their beliefs on the importance of screening [30]. We suggest that such information could be more effectively disseminated as simple health messages endorsed by Brunei's Ministry of Health via website and social media platforms.

Second, we observed high acceptability of HPV self-sampling among our participants. Meta-analyses have indicated strong acceptance of and preference for self-sampling over clinician sampling [31], mainly due to logistical reasons [32]. More than half of our participants who joined HPV self-sampling cited embarrassment or lack of time due to home and/or work commitments as their major reason of not attending Pap test screening. This suggest that providing flexibility to accommodate women's screening method preference [33], such as the option of self-sampling [34], could improve screening uptake.

Thirdly, most of the detected hr-HPV genotypes in our study were non-16/18, with only one out of nine participants tested positive for HPV 16. Although our sample size is small, this result suggests that it may not be accurate to assume HPV 16 or 18 as common hr-HPV genotypes in Brunei, even though this is true in the global context [35]. Other studies have detected a significant percentage of non-16/18 hr-HPV genotypes, suggesting the presence of region-specific heterogeneity in the HPV genotype distribution [35–39]. Also, variation in HPV distribution among different ethnic groups has been reported in an American study [40]. Our finding has potential implication on Brunei's national school-based HPV vaccination program

[41] which currently provides vaccines which do not confer protection against non-16/18 genotypes. Larger population-based studies to understand the distribution of HPV genotypes among Bruneian women will be crucial to determine the efficacy or impact of the current vaccines.

One notable point for the local context relates to the presence of cultural barriers. Being a predominantly Islamic society, religious and cultural modesty could be a contributing factor for embarrassment among Muslim women [42]. Also, having premarital sex is a taboo in Brunei and is generally not openly discussed [43]. This could prevent any unmarried but sexually active women from participating in the screening program. It should be emphasized that only married or ever married females were included in this study; included because it is part of the eligibility criteria for the national cervical cancer screening program in Brunei.

A major limitation for this study is that non-attendees from only one health center were recruited, thus our findings are not representative of the adult female population in Brunei. Secondly, our findings should also be interpreted with caution due to the small sample size and the non-probability sampling approach used. This study was initially conceived as a pilot study due to resource and logistics limitations. There were two reasons for choosing JPSHC as our study site: It is the third largest primary government healthcare center in the country, and that it serves a sub-district with a sizable percentage of residents in the middle- to low-income groups. However, even at this pilot stage, our study findings could encourage stakeholders to conduct similar and larger studies, using random sampling approach. Lastly, as this study relied on the self-reported history of previous Pap test attendance, we might have missed recruiting those who may have forgotten their last Pap test date.

In conclusion, our findings indicate the need to further promote knowledge on cervical cancer, the benefits of screening and clarifying any misconceptions of Pap test results. Reasons of cervical cancer screening non-attendance were mainly related to emotional and logistical factors. As we found high acceptance towards HPV self-sampling, this could be adopted as an alternative for women who refrain from Pap test. Our high detection of non-16/18 HPV genotypes suggest high prevalence of other hr-HPV genotypes in Brunei. Future larger studies involving more Bruneian women should be done to verify our results. Follow-up studies should also be conducted to consider HPV testing as the suitable method for cervical cancer screening.

## Supporting information

**S1 Table. Responses on their major and minor reasons for not attending cervical cancer screening among non-attendees at JPSHC, Brunei (Jan–Dec 2019).**
(DOCX)

**S2 Table. Attitudes towards cervical cancer screening among non-attendees at JPSHC, Brunei (Jan–Dec 2019).** Responses from the total study population (n = 174) were included.
(DOCX)

**S3 Table. Socio demographic characteristics and comparison between screening non-attendees who tested positive and negative for hr-HPV at JPSHC, Brunei (Jan–Dec 2019).**
(DOCX)

**S1 Fig. Preferred sources of information about cervical cancer among non-attendees at JPSHC, Brunei (Jan–Dec 2019).** The x-axis indicates the percentage, and the number next to each bar indicates the number of responses. Multiple responses were allowed and responses from the total study population (n = 174) were included.
(DOCX)

**S1 Questionnaire. Reasons why women do not participate in the national cervical cancer screening program.**
(DOCX)

**S2 Questionnaire. Acceptability of self sampling for HPV testing among non-attendees of cervical screening program in Brunei Darussalam.**
(DOCX)

**S1 Dataset. Minimal dataset.**
(XLSX)

## Acknowledgments

The authors would like to thank the JPSHC staff who have assisted in participant recruitment and questionnaire distribution.

## Author Contributions

**Conceptualization:** Liling Chaw, Edwin Lim, Roslin Sharbawi.

**Data curation:** Liling Chaw, Nurul Iffah Hazwani Ja'afar.

**Formal analysis:** Liling Chaw.

**Funding acquisition:** Edwin Lim, Roslin Sharbawi.

**Investigation:** Nurul Iffah Hazwani Ja'afar, Roslin Sharbawi.

**Methodology:** Liling Chaw, Edwin Lim, Roslin Sharbawi.

**Project administration:** Roslin Sharbawi.

**Resources:** Edwin Lim, Roslin Sharbawi.

**Supervision:** Liling Chaw, Edwin Lim, Roslin Sharbawi.

**Visualization:** Shirley H. F. Lee.

**Writing – original draft:** Liling Chaw, Shirley H. F. Lee, Nurul Iffah Hazwani Ja'afar, Roslin Sharbawi.

**Writing – review & editing:** Liling Chaw, Shirley H. F. Lee, Edwin Lim, Roslin Sharbawi.

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
