## [Decision Letter · Decision Letter 0]

13 Aug 2021

PONE-D-21-02442

Reasons for non-attendance to cervical cancer screening and acceptability of HPV self-sampling among Bruneian women: A cross-sectional study.

PLOS ONE

Dear Dr. Chaw,

Thank you for submitting your manuscript to PLOS ONE. After careful consideration, we feel that it has merit but does not fully meet PLOS ONE’s publication criteria as it currently stands. Therefore, we invite you to submit a revised version of the manuscript that addresses the points raised during the review process.

We look forward to receiving your revised manuscript.

Kind regards,

Oathokwa Nkomazana, MD MSC PhD

Academic Editor

PLOS ONE

Additional Editor Comments (if provided):

This is a very interesting subject.

Please address the comments from reviewers.

Journal Requirements:

Reviewers' comments:

Reviewer's Responses to Questions

**Comments to the Author**

1. Is the manuscript technically sound, and do the data support the conclusions?

Reviewer #1: Partly

Reviewer #2: Partly

2. Has the statistical analysis been performed appropriately and rigorously? 

Reviewer #1: Yes

Reviewer #2: No

3. Have the authors made all data underlying the findings in their manuscript fully available?

Reviewer #1: Yes

Reviewer #2: Yes

4. Is the manuscript presented in an intelligible fashion and written in standard English?

Reviewer #1: Yes

Reviewer #2: No

5. Review Comments to the Author

Reviewer #1: The authors have undertaken a questionnaire based small survey in Brunei women about attitude towards cervical screening. Background is that Brunei has a high incidence of cervical cancer and a low screening coverage of 44%.

The results are worth informing the world about. But the manuscript is fair too long for what it contains. It should be shortened considerably:

1. delete most of page 6

2. Delete figure 2

3. Merge Tables 3 and 4

4. Delete figure 3

5. Shorten considerable the three points in the Discussion – really to about two lines each

I miss Discussion about:

1. why is the mean age so high 45 years?

2. Screening coverage is really not so low in participants: 52.3% had been screening within last 4-10 years

3. How does the reported screening coverage in the group correspond to the national coverage of 44%?

4. By for the majority of women answer agree to “having a pap test taken is beneficial for my health” - when why do the authors recommend more education and awareness as their top priority? - women actually seem to be fairly good informed.

5. More comments should be given on potential practical and cultural barriers.

Overall: relevant topic, but not acceptable for publication in its present form. Should be reduced to a max. Of haft the number of words to be of interest for a broader audience.

Reviewer #2: This is an important topic. Please consider how you can make these results of 174 women generalizable to the entire Brunei population. How do you exclude sampling bias? Why did you do this in two parts?

6. PLOS authors have the option to publish the peer review history of their article (what does this mean?). If published, this will include your full peer review and any attached files.

Reviewer #1: No

Reviewer #2: **Yes: **Diane M Harper

---

## [Author Response · Author response to Decision Letter 0]

24 Aug 2021

Point-to-point responses:

From the Journal Author’s Response

The manuscript has been revised as requested, including the use of American English.

Participants gave written consent for both parts of the study. We have added this point in the Methods section (3rd paragraph under Study design and data collection sub-section):

“Participants who gave written consent to the first part of the study were asked to complete the first questionnaire (Q1) onsite. They were then given an envelope containing information on the second part and an instruction leaflet on the HPV self-sampling procedure. Upon second contact about two weeks later, those who gave verbal consent were given an appointment to JPSHC. On the day, a trained nurse first explained the procedure using an instructional video and answered any questions. Participants who gave another written consent were then given a self-sampling kit and asked to perform the procedure in the clinic. After completion, they were asked to complete the second questionnaire (Q2).”

Both questionnaires used in this study are now attached as Supplementary Information (S4 & S5). 

4. In your Data Availability statement, you have not specified where the minimal data set underlying the results described in your manuscript can be found. PLOS defines a study's minimal data set as the underlying data used to reach the conclusions drawn in the manuscript and any additional data required to replicate the reported study findings in their entirety. All PLOS journals require that the minimal data set be made fully available. 

We have attached the minimal dataset as Supplementary Information (S6). 

Comments from the reviewers Author’s Response

Reviewer #1: The authors have undertaken a questionnaire based small survey in Brunei women about attitude towards cervical screening. Background is that Brunei has a high incidence of cervical cancer and a low screening coverage of 44%.

The results are worth informing the world about. But the manuscript is fair too long for what it contains. It should be shortened considerably:

1. delete most of page 6

2. Delete figure 2

3. Merge Tables 3 and 4

4. Delete figure 3

Thank you for your comments. We have edited the whole manuscript to correct any grammatical mistakes and also make our points shorter and more concise. 

As suggested, we have also:

1. Shorten the data collection method in page 6.

2. Merged Tables 3 and 4, whereby the total count column was added into the original Table 4. The original Table 3 is now moved to the Supplementary Information (as Table S2), for readers who want to refer to the breakdown responses (agree, neutral & disagree).

3. Figure 3 was also moved to the Supplementary Information (as Figure S1), for readers who want to know the full breakdown of responses.

For Figure 2, we decided to keep Figure 2 because we think it is the best way to summarize both counts & percentages for both major and minor reasons (and showing the complete picture of our study results). Table S1 shows the same results in the Table version, but it is less intuitive as Figure 2. Also, Table 2 only show the results for the top 10 major reasons (meaning that it is not showing the complete picture, like Figure 2).

5. Shorten considerable the three points in the Discussion – really to about two lines each

I miss Discussion about:

1. why is the mean age so high 45 years?

2. Screening coverage is really not so low in participants: 52.3% had been screening within last 4-10 years

3. How does the reported screening coverage in the group correspond to the national coverage of 44%?

4. By for the majority of women answer agree to “having a pap test taken is beneficial for my health” - when why do the authors recommend more education and awareness as their top priority? - women actually seem to be fairly good informed.

5. More comments should be given on potential practical and cultural barriers.

5.1. The median age of 45 years could be due to the specific population that we have targeted, that is, the screening non-attendees. As mentioned in the Methods section: “The inclusion criteria were married or ever married Bruneian women between 20 and 65 years old, who either never or did not have a Pap test within the last 4 years, and who can comprehend Brunei-Malay or English language”.

While it is possible this could be due the convenience sampling that were conducted in this study, we take note that previous studies from Finland (Virtanen A et al. Ref no.18 in manuscript) and El Salvador (Maza M, et al. Ref no.5 in manuscript), both focusing on screening non-attendees only, also have similar median ages. Although both studies include women between 30-60 years old, our study is still comparable with theirs as the median age of marriage in Brunei is about 26 years from 2014 to 2019 (based on country census data: http://www.deps.gov.bn/SitePages/Vital%20Statistics.aspx )

In the revised manuscript, we decided not to explicitly include this point. Instead, we have added a new paragraph on the potential cultural barriers (please see item 5.5 below) which elude to this point.

5.2. Thank you for your point here. In this study, we want to investigate reasons for both non-attendance to screening at all, as well as for not continuing with screening every 3 years, as recommended. This is why women who had undergone screening within the last 4-10 years were included in this study. To clarify this point and avoid confusion, we have replaced the word “coverage” to “uptake” in our study significance statement (Introduction section, last sentence): “Study findings could be used to strategize ways to improve screening uptake and provide preliminary evidence towards implementing HPV testing as the primary screening test for cervical cancer in Brunei.”

5.3. This point relates to the sampling strategy for this study. Briefly, our study findings could not be generalizable to the whole country population. Please see our response for Reviewer #2 below, who commented on the result generalizability. 

5.4. We would like to clarify that we are pushing for promoting knowledge on cervical cancer, the benefits of screening and clarifying any misconceptions of Pap test results. This implies not only on promoting general awareness on the Pap test, but also on why is this important to be conducted periodically and to clarify any misconceptions of the Pap test results. To prevent any misunderstanding, the conclusion paragraph is revised as follows: 

“In conclusion, our findings indicate the need to further promote knowledge on cervical cancer, the benefits of screening and clarifying any misconceptions of Pap test results. Reasons of cervical cancer screening non-attendance were mainly related to emotional and logistical factors. As we found high acceptance towards HPV self-sampling, this could be adopted as an alternative for women who refrain from Pap test. Our high detection of non-16/18 HPV genotypes suggest high prevalence of other hr-HPV genotypes in Brunei. Future larger studies involving more Bruneian women should be done to verify our results. Follow-up studies should also be conducted to consider HPV testing as the suitable method for cervical cancer screening.”

5.5. Thank you for your point here. We have added a new paragraph on the potential cultural barriers: 

“One notable point for the local context relates to the presence of cultural barriers. Being a predominantly Islamic society, religious and cultural modesty could be a contributing factor for embarrassment among Muslim women [39]. Also, having premarital sex is a cultural taboo in Brunei and is generally not openly discussed [40]. This could prevent any unmarried but sexually active women from participating in the screening program. It should be emphasized that only married or ever married females were included in this study; included because it is part of the eligibility criteria for the national cervical cancer screening program in Brunei.”

Overall: relevant topic, but not acceptable for publication in its present form. Should be reduced to a max. Of haft the number of words to be of interest for a broader audience.

Thank you for your comment. We have edited the manuscript and have reduced the word count to 3049 words (manuscript, excluding references). We hope this revision would be clear and concise enough for the journal readers.

Reviewer #2: This is an important topic. Please consider how you can make these results of 174 women generalizable to the entire Brunei population. How do you exclude sampling bias? Why did you do this in two parts?

Thank you for your comments.

(a) and (b). It will be difficult to generalize the study results, as there was only one study site, sample size was low & sampling bias was done conveniently. We have emphasized this in the limitations: 

“A major limitation for this study is that non-attendees from only one health center were recruited, thus our findings are not representative of the adult female population in Brunei. Secondly, our findings should also be interpreted with caution due to the small sample size and the non-probability sampling approach used. This study was initially conceived as a preliminary study due to resource and logistics limitations. There were two reasons for choosing JPSHC as our study site: It is the third largest primary government healthcare center in the country, and that it serves a sub-district with a sizable percentage of residents in the middle- to low-income groups. It is hoped that even at this preliminary stage, our study findings could encourage stakeholders to conduct similar and larger studies, using random sampling approach.”

(c) Thank you for pointing this out. This study was originally carried out as a two-part study (one on the reasons for non-attendance questionnaire & another on self-sampling). However, we understand that it is irrelevant to mention this, and have therefore removed it entirely from the manuscript: in abstract (first sentence in Methods), and in discussion (first sentence).

---

## [Decision Letter · Decision Letter 1]

8 Nov 2021

PONE-D-21-02442R1Reasons for non-attendance to cervical cancer screening and acceptability of HPV self-sampling among Bruneian women: A cross-sectional study.PLOS ONE

Dear Dr. Chaw,

Thank you for submitting your manuscript to PLOS ONE. After careful consideration, we feel that it has merit but does not fully meet PLOS ONE’s publication criteria as it currently stands. Therefore, we invite you to submit a revised version of the manuscript that addresses the points raised during the review process.

Thank you for reporting on your important work.

 Please attend to all the comments by third reviewer. It may be helpful to include, in the method section, "The context of the study". Under this section, briefly outline the national cervical cancer guideline. At what age is screening started? How often should screening be done? What is the screening method? And any other information that will help the reader situate the problem you are addressing through this study.

We look forward to receiving your revised manuscript.

Kind regards,

Oathokwa Nkomazana, MD MSC PhD

Academic Editor

PLOS ONE

Journal Requirements:

Additional Editor Comments:

Thank you for reporting on your important work.

Please attend to all the comments by third reviewer. It may be helpful to include, in the method section, "The context of the study". Under this section, briefly outline the national cervical cancer guideline. At what age is screening started? How often should screening be done? What is the screening method? And any other information that will help the reader situate the problem you are addressing through this study.

Reviewers' comments:

Reviewer's Responses to Questions

**Comments to the Author**

1. If the authors have adequately addressed your comments raised in a previous round of review and you feel that this manuscript is now acceptable for publication, you may indicate that here to bypass the “Comments to the Author” section, enter your conflict of interest statement in the “Confidential to Editor” section, and submit your "Accept" recommendation.

Reviewer #2: All comments have been addressed

Reviewer #3: (No Response)

2. Is the manuscript technically sound, and do the data support the conclusions?

Reviewer #2: Yes

Reviewer #3: No

3. Has the statistical analysis been performed appropriately and rigorously? 

Reviewer #2: Yes

Reviewer #3: I Don't Know

4. Have the authors made all data underlying the findings in their manuscript fully available?

Reviewer #2: Yes

Reviewer #3: Yes

5. Is the manuscript presented in an intelligible fashion and written in standard English?

Reviewer #2: Yes

Reviewer #3: Yes

6. Review Comments to the Author

Reviewer #2: great work! Continue to probe why older women will not screen after you do an educational intervention that shows the average age of cx ca is about 40 years old without any symptoms.

Reviewer #3: General comments:

Piloting self-testing in various cultural and geographic settings is important, however, I do not believe this adds to the body of literature supporting HPV self-testing. The sample size is also relatively small to make any meaningful comments about HPV prevalence; the authors do cite the need for additional research. The scope of this work would be more appropriate for a regional journal.

Non-attendees needs a clear definition somewhere, as well as clarification as to why a Pap test > 4 years ago was included (is this the recommendation in Brunei?).

Throughout the paper, you contrast HPV self swabs to a Pap test. I think the comparison you are really making is between a self-swab and a provider-collected test (using a pelvic examination). Or do you think there is a patient held belief in the reliability of a Pap test? Do you think it would be appropriate to rather compare HPV testing to any cervical cancer screening? Do you think there was any role of the desire to have a pelvic examination, other contexts have shown that patients feel more secure when a doctor as examined them.

Specific comments:

Abstract:

Objective: specify <80% where? Explain non-attendance – do you mean non-attendance at recommended screening/screening intervals?

Methods: describe the health center – tertiary referral center? You mention 4 years – is this the interval for screening in Brunei? You don’t mention the second survey here. Statistical analysis is not sufficient.

Results: it is hard for a reader to understand how you enrolled “non-attendees”. Perhaps that can simply be clarified. Need to specify in line 46 how many women accepted HPV self-swab.

Discussion: In the results you state that “Fear of bad results, embarrassment and lack of time” are the main causes for non-attendance, I’m not sure your suggested intervention directly tackles those issues. Perhaps specific education/community awareness that cervical cancer is a preventable disease? Line 52 is non-specific and needs clarification.

Introduction: Generally not linear in its progression. First paragraph starts with effectiveness of screening and in particular screening with HPV testing. Second paragraph starts with more effective screening strategies for screening non-attendees. Third paragraph with specifies to Brunei. These are fine themes, but the authors lose focus as they add to these paragraphs. Also, I think there should be a more in-depth dive into screening non-attendance. What are patterns internationally and/or what are factors that cause non-attendance. Specific feedback:

Line 69 and 71: Screening non-attendees: what does this mean. How do you survey screening non-attendees? I assume there is a definition in methods, but some explanation is needed here.

Lines 73-74: need full definition for CIN2+, you have only spelled out CIN

Lines 74-79 seem to belong in the line of thinking of the first paragraph

Lines 89-91 seem out of place and belong in the second paragraph

Lines 95-97 make the statement about the importance/impact of this study more definitive.

Methods: The methods should be clear enough for someone else to repeat the study. It is not clear who was collecting the data, how issues of illiteracy were addressed (or was this an exclusion criteria). It is not stated where data was stored and analysis is not clear enough to be replicable.

Perhaps call this a pilot instead of “preliminary”

Line 115: not clear what “first part of the study is”

Line 117: how was second contact attempted? Why two weeks? I would suggest incorporating what was on the questionnaires with the first reference to the questionnaire, rather than having it as another sectioin

Line 131: could not can

Line 173: It is not clear if frequencies were used to present the questions that were rated on a likert scale for the classifications described. If so, how were these frequencies compared (I notice the p-value in the results below).

Results:

Line 195: drop “major” and “minor” qualifiers or define them.

Line 219: what does this mean they agreed on the “severity of cervical cancer”?

Line 234, would avoid the use of subjective qualifiers

Line 248, It is pretty remarkable that 8/9 women with HPV had abnormal pap smears. Also does this mean only 1 participant had CIN3? What about the rest?

Discussion:

First sentence can be eliminated.

Line 310 – this should be mentioned in the methods section in exclusion criteria of non-married women

Lines 272-274 – this hadn’t come up in the results, so I wouldn’t bring it into the discussion

Lines 319 – 321 – rephrase this, don’t use “hope” to convey this sentiment

Tables:

Table 1: percentages are confusing because you don’t know how many women are in each age group, so you don’t know if the percentage is of the total number of women or the number in the age group. I think the percentages would be more understandable if they were the percentage of women in the age group who joined/declined. What’s the difference between “never” and “no” for HPV vaccination?

Table 4, question 12 – do you think the wording of the question was leading to participants by putting “proper” before Pap test?

7. PLOS authors have the option to publish the peer review history of their article (what does this mean?). If published, this will include your full peer review and any attached files.

Reviewer #2: **Yes: **Diane M Harper

Reviewer #3: No

---

## [Author Response · Author response to Decision Letter 1]

30 Nov 2021

Please see attached the point-by-point response for each comment.

---

## [Editor Report · Decision Letter 2]

20 Dec 2021

Reasons for non-attendance to cervical cancer screening and acceptability of HPV self-sampling among Bruneian women: A cross-sectional study.

PONE-D-21-02442R2

Dear Dr. Chaw,

We’re pleased to inform you that your manuscript has been judged scientifically suitable for publication and will be formally accepted for publication once it meets all outstanding technical requirements. Please address two minor issues: 1. Line 77: indicate the unit and period of the incidence rate. Is the unit percentage or per thousand population? Is the period per year or what?

2.Line 85; first mention of Human Papilloma Virus shouldn't use acronym only.

3. Instead of calling your participants non-attendees, it may be better to use a descriptive phrase like: 'those currently not accessing screening'

Kind regards,

Oathokwa Nkomazana, MD MSC PhD

Academic Editor

PLOS ONE
---

## [Editor Report · Acceptance letter]

5 Mar 2022

PONE-D-21-02442R2 

Reasons for non-attendance to cervical cancer screening and acceptability of HPV self-sampling among Bruneian women: A cross-sectional study. 

Dear Dr. Chaw:

I'm pleased to inform you that your manuscript has been deemed suitable for publication in PLOS ONE. Congratulations! Your manuscript is now with our production department. 

Kind regards, 

on behalf of

Dr. Oathokwa Nkomazana 

Academic Editor

PLOS ONE